
# Entanglement-enabled symmetry-breaking orders

**Cheng-Ju Lin[1,2,3] and Liujun Zou[1]**

**1** Perimeter Institute for Theoretical Physics, Waterloo, Ontario N2L 2Y5, Canada
**2** Joint Center for Quantum Information and Computer Science,
NIST/University of Maryland, College Park, Maryland 20742, USA
**3** Joint Quantum Institute, NIST/University of Maryland, College Park, Maryland 20742, USA

## Abstract

A spontaneous symmetry-breaking order is conventionally described by a tensor-product wavefunction of some few-body clusters; some standard examples include the simplest ferromagnets and valence bond solids. We discuss a type of symmetry-breaking orders, dubbed *entanglement-enabled symmetry-breaking orders*, which *cannot* be realized by any such tensor-product state. Given a symmetry breaking pattern, we propose a criterion to diagnose if the symmetry-breaking order is entanglement-enabled, by examining the compatibility between the symmetries and the tensor-product description. For concreteness, we present an infinite family of exactly solvable gapped models on one-dimensional lattices with nearest-neighbor interactions, whose ground states exhibit entanglement-enabled symmetry-breaking orders from a discrete symmetry breaking. In addition, these ground states have gapless edge modes protected by the unbroken symmetries. We also propose a construction to realize entanglement-enabled symmetry-breaking orders with spontaneously broken continuous symmetries. Under the unbroken symmetries, some of our examples can be viewed as symmetry-protected topological states that are beyond the conventional classifications.

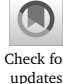

# 1 Introduction

Spontaneous symmetry breaking is a ubiquitous phenomenon and plays fundamental roles in numerous physical systems, ranging from astronomical objects like neutron stars to quantum mechanical objects like atoms forming a crystal. It is also the basis of many important modern technologies, such as hard drives, maglev trains, and spintronics.

In quantum physics, a standard recipe in characterizing a spontaneous symmetry breaking order is by representing it as a tensor-product wavefunction of some few-body clusters. For example, as depicted in Fig 1, a quantum ferromagnetic order is often described as a tensor product of spin-ups or spin-downs; a valence-bond-solid order is described as a tensor product of spin-singlets. Note that while a generic symmetry-breaking state is not a tensor-product wavefunction, here we stress that the recipe is about the *possibility* of realizing such a symmetry breaking state via a tensor-product wavefunction as a characteristic description. The tensor-product description is also usually the "fixed point" wavefunction of the symmetry breaking orders. This conventional wisdom seems to suggest that all symmetry breaking orders can be represented by a tensor-product wavefunction.

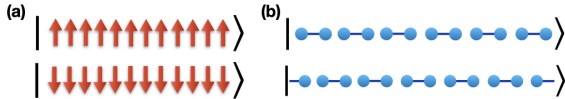

Figure 1: "Classical" symmetry-breaking orders. (a) A quantum ferromagnetic order. (b) A valence-bond-solid order.

Nevertheless, remarkably, there exist symmetry-breaking orders that are *impossible* to be described by any tensor-product state, fundamentally requiring entanglement. We dub such orders *entanglement-enabled symmetry-breaking orders* (EESBOs), which will be defined more precisely later. How would such orders arise? An expected scenario is where the remaining symmetry has some Lieb-Schultz-Mattis (LSM) constraints, mandating the states to be long-range entangled. Later we will review how LSM constraints from the remaining symmetry mandate EESBOs via an example from Ref. [1].

On the other hand, in this work, we investigate a less anticipated and less discussed situation where the symmetries do not have LSM constraints but are still incompatible with any tensor-product wavefunctions. That is, although the wavefunction is short-range entangled, it can never be smoothly deformed to a limit represented by a tensor-product wavefunction, as long as the given symmetry-breaking pattern is to be realized. We make this notion of EESBO precise, and propose a diagnostic to check if a symmetry-breaking order is entanglement-enabled. To illustrate the idea more concretely, we construct models with exactly solvable ground states to unambiguously prove that such EESBOs can exist, where the spontaneously broken symmetries are discrete. We then discuss more examples with spontaneously broken continuous symmetries, which show additional interesting properties, such as novel structures of the Goldstone modes.

In passing, we note that models exhibiting short-range EESBOs which can be diagnosed by our diagnostic have appeared in Refs. [2–4]. However, the motivations to study such models appear to not be based on EESBOs, and the obstruction of the local-product-state realization of the order was not shown explicitly therein. In Appendix A, we will apply our diagnostic to verify that they are indeed EESBOs.

Here we point out some potential significance of EESBO. The usual description of the symmetry-breaking orders suggests that all symmetry-breaking orders are essentially "classical" [5], since they allow a description by a tensor-product wavefunction. On the other hand, EESBO is intrinsically quantum, indicating the incompleteness of the above description. For instance, recent development in the systematic understanding of the magnon topological band structures is based on "classical" symmetry-breaking orders [6–10]. However, the complete understanding must incorporate the magnons arising from EESBOs. Furthermore, the properties of the symmetry defects associated with EESBOs are largely unexplored and could pave a way to future technologies, much like the roles of skyrmions in spintronics [11–13].

This paper is organized as follows. In Sec. 2, we present a precise definition of EESBOs, which is followed by some convenient diagnostics to check whether a symmetry-breaking order is entanglement-enabled in Sec. 3. In Sec. 4, we provide an infinite family of models with exactly solvable ground states that exhibit EESBOs (together with some other interesting properties), where the spontaneously broken symmetries are discrete. In Sec. 5, we discuss more examples of EESBOs where the spontaneously broken symmetries are continuous. We finish the paper with discussion in Sec. 6. Additional details are presented in various appendices.

## 2 Definition of entanglement-enabled symmetry-breaking orders

To define EESBOs precisely, we first need to clarify how to specify the symmetry setting of a physical system, and the notion of a tensor-product state of few-body clusters. We will see that EESBOs arise from some incompatibility between the symmetries and tensor-product wavefunctions.

For concreteness, we consider a quantum system on a lattice, described by a tensor product of local Hilbert spaces. Changing the structures of the local Hilbert spaces, i.e., their dimensions and operator contents, changes the physical system. Below we often use "spin-$S$ system" to mean a bosonic system with $2S + 1$ dimensional local Hilbert spaces. We assume that the system is described by some local Hamiltonian with a symmetry group $G_0$.

To describe spontaneous symmetry breaking, we first have to specify the symmetries of the Hamiltonian.[1] In particular, we need the symmetry setting specified by i) the group $G_0$ formed by the symmetry actions on *local operators*, ii) how *states in each local Hilbert space* transform

---

[1]We only consider 0-form, ordinary symmetries.

under the symmetry, and iii) the locations of the degrees of freedom (DOF).[2] For example, a spin system on a kagome or honeycomb lattice described by the usual Heisenberg Hamiltonian has $G_0 = SO(3) \times \mathbb{Z}_2^T \times p6m$, where $SO(3)$, $\mathbb{Z}_2^T$ and $p6m$ describe the spin rotation, time reversal and lattice symmetries, respectively. We stress that the group $G_0$ does not fully specify this symmetry setting without specifying how each spin transforms under $G_0$, e.g., the magnitude of the spin, whether it is a Kramers doublet, etc. We also need to specify the locations of the spins, i.e., whether they live on a kagome or honeycomb lattice. However, for convenience, we just use the group $G_0$ to denote the symmetry setting, keeping other information implicit.

The ground states of the system may spontaneously break the symmetry $G_0$ to its subgroup $G$,[3] and we denote this symmetry-breaking order by "$G_0 \to G$". To define EESBO more formally, let us define the "classical" symmetry-breaking order first.

**Definition 1** *Given $G_0$ and $G$, we call the symmetry-breaking order "classical" if its G-symmetric ground states $|\psi\rangle$ can be represented by a local product state, i.e., $|\psi\rangle = \bigotimes_i |\psi_i\rangle_{\Lambda_i}$, where $|\psi_i\rangle_{\Lambda_i}$ is supported only in a local region $\Lambda_i$ for all i.*

Here by local we mean that the "size" of $\Lambda_i$ does not scale with the system size, and we will call the state "ultralocal" if each $\Lambda_i$ is just one site. Note that although a "classical" $G_0 \to G$ symmetry-breaking order can be realized by local product states, generically it can also be realized by other states, which may have (even long-range) entanglement, since $G$ alone does not fully determine the ground states. As mentioned in Sec. 1 and Fig. 1, the all-up or all-down ferromagnetic states and the valance-bond-solid state are examples of the classical order, since the $G$-symmetric states (i.e., , the $G_0$ symmetry-breaking states) can be realized as a local tensor product state.

We now give a definition of EESBO, which is defined as the non-"classical" symmetry breaking orders.

**Definition 2** *Given the symmetry breaking relation $G_0 \to G$, if all possible G-symmetric ground states cannot be realized by local product states but can be realized by some entangled states, we call the $G_0 \to G$ symmetry-breaking order an "entanglement enabled symmetry-breaking order" (EESBO).*

It is worth emphasizing that the above definition of EESBO is different from the usual phenomenon of coexistence of spontaneous symmetry breaking and nontrivial topological phase. In the latter, the ground state with spontaneously broken symmetries can be a nontrivial topological phase, but usually one assumes that it does not have to be a nontrivial topological phase and the symmetry breaking pattern of interest can be represented by a local product state. However, in our definition of EESBO, the ground states with the relevant symmetry breaking pattern can never be represented by any local product state. An explicit example that highlights this difference is given at the end of Sec. 4.

## 3 Diagnostics for long-range and short-range EESBOs

Having defined EESBOs, in this section we provide some convenient diagnostics that can be used to check whether a symmetry-breakng order is entanglement-enabled.

---

[2]In this definition of a symmetry setting, if there is a $U(1)$ symmetry, the filling factor is allowed to be tuned after specifying these three pieces of data, unless it is pinned by other symmetries.

[3]How states transform under $G$ is determined by how they transform under $G_0$, and the locations of the DOF are often unchanged after spontaneous symmetry breaking. So the symmetry setting after spontaneous symmetry breaking is fully specified.

As discussed in the Introduction, a somewhat expected scenario that gives rise to an EESBO is if there are LSM-type constraints[4] that mandate all $G$-symmetric ground states to be long-range entangled [14–17], so the corresponding symmetry-breaking order must be entanglement-enabled. For instance, consider a kagome lattice qubit system with $G_0 = SO(3) \times \mathbb{Z}_2^T \times p6m$, i.e., $G_0$ includes an $SO(3)$ spin rotation symmetry where the qubits carry spin-1/2, a $\mathbb{Z}_2^T$ time reversal symmetry where the qubits transform as Kramers doublets, and a $p6m$ lattice symmetry that moves the locations of the qubits. It is proposed that $G_0 \to G = SO(3) \times p6m$ for some Hamiltonian [1] (i.e., $\mathbb{Z}_2^T$ is spontaneously broken), which has an LSM constraint mandating all $G$-symmetric ground states to be long-range entangled. LSM constraints therefore serve as a useful diagnostic for the long-range entangled type of EESBOs.

Although interesting by itself, the entanglement-enabled nature of such symmetry-breaking orders is anticipated due to the LSM constraints. It may be more remarkable that EESBOs can arise even if $G$-symmetric states have *no* LSM constraint, i.e., $G$-symmetric states can be short-range entangled but *cannot* be local product states. Given the original and remaining symmetries $G_0$ and $G$ without the LSM constraints, how do we diagnose if the symmetry-breaking order is entanglement-enabled? Assuming the $G_0 \to G$ symmetry-breaking order is realizable, we can have a useful general criterion to show its entanglement-enabled nature: $G$ contains a lattice symmetry which constrains the structure of any local product state, and an on-site symmetry which is incompatible with this constraint. This symmetry $G$ therefore forbids the possibility of local-product-state realization, forcing the symmetry-breaking order to be entanglement-enabled. In practice, the lattice symmetry will often constrain any local product state to be ultralocal, which is then incompatible with the on-site symmetry.

## 4 EESBOs with a broken discrete symmetry

We first show an infinite family of examples of EESBO with a broken discrete symmetry. These examples are inspired by a setup in Ref. [18], and we will point out the difference between our consideration and Ref. [18] later. Each of this infinite family of EESBOs lives on a one dimensional lattice, where each site hosts an $n^2$ dimensional local Hilbert space, labeled by an integer $n \geqslant 3$. We represent a basis of the local Hilbert space at site $j$ by $|\alpha, \beta\rangle_j$, where $\alpha, \beta = 0, 1, \cdots, n-1$ are defined modulo $n$. We consider the generalized Pauli operators $\mu_j^{x,z}$ and $\nu_j^{x,z}$, where

$$\mu_j^z |\alpha, \beta\rangle_j = e^{\frac{2\pi i}{n}\alpha} |\alpha, \beta\rangle_j, \qquad \mu_j^x |\alpha, \beta\rangle_j = |\alpha+1, \beta\rangle_j,$$

$$\nu_j^z |\alpha, \beta\rangle_j = e^{\frac{2\pi i}{n}\beta} |\alpha, \beta\rangle_j, \qquad \nu_j^x |\alpha, \beta\rangle_j = |\alpha, \beta+1\rangle_j.$$

While it is convenient to view variables $\alpha$ and $\beta$ as *fictitious* DOF, it is crucial to keep in mind that the local Hilbert space is formed by $\alpha$ and $\beta$ *together*; in particular, one cannot discuss the entanglement between the $\alpha$ and $\beta$ DOF at the same site.

The symmetry group $G_0$ of the Hamiltonian contains a $\mathbb{Z}_n^x \times \mathbb{Z}_n^z$ internal symmetry, where $\mathbb{Z}_n^x$ is implemented by $\prod_j \mu_j^x \nu_j^x$, and $\mathbb{Z}_n^z$ is implemented by $\prod_{j \in \text{even}}(\mu_j^z \nu_j^z) \prod_{j \in \text{odd}}(\mu_j^z \nu_j^z)^{-1}$. $G_0$ also contains a lattice translation symmetry $T: \bigotimes_j |\alpha_j \beta_j\rangle_j \to \bigotimes_j |\alpha_j \beta_j\rangle_{j+1}$ and a reflection symmetry with respect to the site $j = 0$, $\sigma: \bigotimes_j |\alpha_j \beta_j\rangle_j \to \bigotimes_j |\beta_j \alpha_j\rangle_{-j}$.

We comment on some noteworthy aspects of this symmetry setting. First, while the symmetry $G_0$ can be defined on an infinite 1d lattice, defining it on a finite lattice with $L$ sites

---

[4]As commented before, in our definition the $U(1)$ filling factor is not fixed unless it is pinned by other symmetries, so there is no LSM constraint purely associated with the filling factor.

requires a periodic boundary condition and $L$ to be even. Second, all operators transform in a linear representation of the $\mathbb{Z}_n^x \times \mathbb{Z}_n^z$ symmetry. But the states at each site transform in its projective representations, and pairs of adjacent sites together form a linear representation [18]. Third, $T$ and $\mathbb{Z}_n^x \times \mathbb{Z}_n^z$ do not commute, and our setting has *no* LSM constraint.

## 4.1  Application of the EESBO diagnostic

We now consider the symmetry breaking pattern where $T$ is broken to $T^2$ with other symmetries intact, namely, $G_0 \to G$, where $G$ is generated by $\mathbb{Z}_n^x \times \mathbb{Z}_n^z$, $\sigma$ and $T^2$. Below we show the obstruction in realizing such a symmetry-breaking order as a local product state using the diagnostic laid out previously in Sec. 3 for the short-ranged entangled EESBOs.

We first show that the lattice symmetries $T^2$ and $\sigma$ enforces any local tensor-product description to be ultralocal. Recall that a local tensor-product description $|\psi\rangle = \bigotimes_i |\psi_i\rangle_{\Lambda_i}$ requires the possibly entangled cluster $|\psi_i\rangle_{\Lambda_i}$ to be supported only in a *local* region $\Lambda_i$ which does not scale with the system size for all $i$. If we assume sites $2j_1$ and $2j_2$ (or sites $2j_1 + 1$ and $2j_2 + 1$) are entangled, then $T^2$ would require extensively many sites to be entangled, contradicting the assumption of the local tensor-product description. That is to say, to be representable by a local product state, no lattice sites labeled with even (odd) numbers can be entangled. Now if sites $2j_1$ and $2j_2 + 1$ are entangled, then $\sigma$ requires sites $2j_1$ and $4j_1 - 2j_2 - 1$ to be entangled. $T^2$ further requires sites $4j_1 - 2j_2 - 1$ and $6j_1 - 4j_2 - 2$ to be entangled, and we now see that two even-numbered sites have to be entangled, which would result in a contradiction again. So any tensor-product description must be ultralocal, i.e., the size of $\Lambda_i$ is one. However, because the states at each site transform in projective representations of the $\mathbb{Z}_n^x \times \mathbb{Z}_n^z$ symmetry (recall $n \geqslant 3$), no ultralocal product state is invariant under $\mathbb{Z}_n^x \times \mathbb{Z}_n^z$.

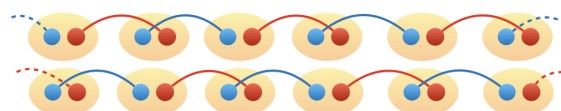

Figure 2: The symmetry-breaking ground states of the Hamiltonian Eq. (1) with periodic boundary condition. With open boundary condition, the ground states have dangling edge degrees of freedom, depicted by the same figure but with the dashed lines removed.

## 4.2  An infinite family of exactly solvable models

We have shown that if this symmetry-breaking order can be realized, they must be EESBOs since it is impossible to be realized as a tensor-product state. In the following, we present explicit parent Hamiltonians and ground states for them, which firmly establishes that such EESBOs are theoretically possible.

Consider the Hamiltonian on a 1d lattice with an even number of sites $L$ and a periodic boundary condition ($j + L \equiv j$),

$$H = \sum_{j=1}^{L} (I - P_{j,j+1}) \,, \tag{1}$$

where $I$ is the identity operator, $P_{j,j+1}$ is the projector onto the linear space

$$\text{span}\{|D_{a,b}^{(\alpha)}\rangle, |D_{a,b}^{(\beta)}\rangle, a, b = 1 \dots n\} \,,$$

where

$$|D_{a,b}^{(\alpha)}\rangle = \sum_{d=1}^{n}|\alpha_j=d,\beta_j=a,\alpha_{j+1}=d,\beta_{j+1}=b\rangle,$$

$$|D_{a,b}^{(\beta)}\rangle = \sum_{d=1}^{n}|\alpha_j=a,\beta_j=d,\alpha_{j+1}=b,\beta_{j+1}=d\rangle,$$

are the "dimers" of the DOF $\alpha$ or $\beta$. It is easy to verify that the two unnormalized zero-correlation-length $G$-symmetric states

$$|\psi_A\rangle = \sum_{\{\alpha_j,\beta_j\}}\prod_{j=1}^{L/2}\delta_{\alpha_{2j}\alpha_{2j+1}}\delta_{\beta_{2j-1}\beta_{2j}}|\alpha_1\beta_1\ldots\alpha_L\beta_L\rangle,$$

$$|\psi_B\rangle = \sum_{\{\alpha_j,\beta_j\}}\prod_{j=1}^{L/2}\delta_{\alpha_{2j-1}\alpha_{2j}}\delta_{\beta_{2j}\beta_{2j+1}}|\alpha_1\beta_1\ldots\alpha_L\beta_L\rangle, \qquad (2)$$

are the ground states, i.e., $H|\psi_{A(B)}\rangle = 0$, which are depicted in Fig. 2. In Appendix B, we show that these two states are indeed the only two ground states of $H$, and that $H$ has a spectral gap in the thermodynamic limit. Note that these two states are orthogonal only in the thermodynamic limit but are linearly independent at any finite $L$.

Intriguingly, these symmetry-breaking ground states are $\mathbb{Z}_n^x \times \mathbb{Z}_n^z$ symmetry-protected topological (SPT) states. This can be shown by examining the ground states with open boundary condition. In this case, the dimension of the ground state subspace becomes $2n^2$, where the extra degeneracy comes from the edge states of the dangling $\alpha$ or $\beta$ DOF, as depicted in Fig. 2 with the dashed lines removed. These edge DOF transform projectively under $\mathbb{Z}_n^x \times \mathbb{Z}_n^z$, which is a hallmark of 1d $\mathbb{Z}_n^x \times \mathbb{Z}_n^z$ SPTs [19–21]. More interestingly, under the remaining $\mathbb{Z}_n^x \times \mathbb{Z}_n^z \times \sigma$ symmetry, these ground states can be viewed as SPTs beyond the conventional group-cohomology-based classification [22, 23], which is possible because the states at each site transform projectively under the on-site symmetries [18]. To capture such states, a more refined classification like the ones in Refs. [18, 24] is needed. We leave the derivation of these statements in Appendix B.

It is worth mentioning that the model analogous to Eq. (1) can be defined for $n = 2$. In this case, similar analysis shows that the ground states still realize a $G_0 \to G$ symmetry-breaking order, and that they are still nontrivial SPTs under the remaining $\mathbb{Z}_2^x \times \mathbb{Z}_2^z$ symmetry. However, the $G_0 \to G$ symmetry-breaking order in this case is not entanglement-enabled, because it can be realized by a tensor-product state, e.g., $\prod_{j\in\text{even}}(|11\rangle_j + |22\rangle_j)\prod_{j\in\text{odd}}(|11\rangle_j - |22\rangle_j)$. This example highlights the difference between the notion of EESBOs and the usual notion of spontaneous symmetry breaking coexisting with nontrivial SPTs.

Before finishing this section, we comment on some differences between the examples discussed above and the one in Ref. [18]. One of the major differences is that our setups have spontaneous symmetry breaking, while the one in Ref. [18] does not. Moreover, in Ref. [18], it was argued that a $G$-symmetric short-range entangled ground state must have nontrivial edge states for the case of $n = 4$. Although it is not explicitly discussed there, this statement is actually true for all $n \geqslant 3$, i.e., for the entire infinite family discussed here. However, in our diagnostic of EESBO above, we actually do not need to and did not use this property.

## 5  EESBOs with spontaneously broken continuous symmetries

The above section firmly establishes that EESBOs can exist even without LSM constraints associated with the remaining symmetry $G$. In this section, we present examples of entanglement-

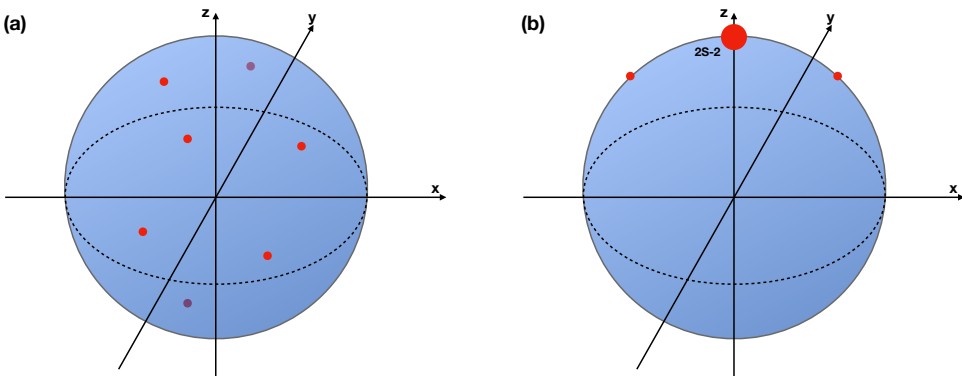

Figure 3: (a) A spin-$S$ tate can be represented as $2S$ points on a Bloch sphere, where the $SO(3)$ rotation can be viewed as a rigid body rotation of the $2S$ points, here $S = 7/2$. (b) An example of a spin-$S$ state which breaks $SO(3)$ symmetry to $\mathbb{Z}_2$, generated by the $\pi$ rotation with respect to $S_z$.

enabled spontaneously broken *continuous* symmetries, which result in richer structures and illustrate some important features of EESBOs.

Consider a $d$-dimensional spin-$S$ system with $G_0 = SO(3) \times \mathbb{Z}^d$, where the $\mathbb{Z}^d$ translation symmetric lattice has one site per unit cell, and a spin-$S$ moment lives on each site. We consider the case where $d > 1$ and $G = \mathbb{Z}_2 \times \mathbb{Z}^d$, i.e., $SO(3)$ is broken to $\mathbb{Z}_2$ while the translation is intact.[5] Note that there is no LSM constraint associated with the symmetry $G$, so we will apply our short-ranged EESBO diagnostic below.

## 5.1 "Classical" symmetry-breaking orders for $S \geqslant 1$

First, it is easy to verify that the translation symmetry forces all local product states to be ultralocal, i.e., $|\psi\rangle = \bigotimes_i |\chi\rangle_i$ where $i$ is the site label and $|\chi\rangle$ is a spin-$S$ wavefunction. It therefore suffices to examine the compatibility of the onsite $SO(3) \to \mathbb{Z}_2$ symmetry with the tensor-product description.

To start, first recall that any spin-$S$ state can be uniquely represented by $2S$ points on a Bloch sphere [25–27] (see Fig. 3(a)); this representation is known as the Majorana representation, and it is a generalization of the familiar statement that any spin-1/2 state can be uniquely represented by a single point on a Bloch sphere. The validity of this representation is based on the fact that all spin-$S$ states can be constructed by symmetrizing $2S$ spin-1/2 states. The advantage of the Majorana representation is that the $SO(3)$ spin rotation acts as a rigid body rotation of these $2S$ points.

Now it is easy to see that for all $S \geqslant 1$, there are single-site spin-$S$ states $|\chi\rangle$ with a $\mathbb{Z}_2$ symmetry generated by the $\pi$ rotation with respect to $S_z$, and one example is to put 2 of the $2S$ points at spherical coordinates with polar angle anything but not $\pi/2$, and azimuthal angles $0$ and $\pi$ for each of them. The other $2S-2$ points are put on the north pole (see Fig. 3(b)). It is easy to check that such a state has no symmetry other than the $\mathbb{Z}_2$. Therefore, the $G_0 \to G$ symmetry-breaking order can be represented by an ultralocal product state for $S \geqslant 1$, and a representative wavefunction is simply taking the spin at each site to be the $\mathbb{Z}_2$ symmetric single-site state discussed here.

On the other hand, if $S = 1/2$, all translation symmetric ultralocal product states have a remaining $U(1)$ on-site symmetry. Specifically, any translation symmetric ultralocal product state of a spin-1/2 system has a wavefunction of the form $\bigotimes_i \left( \cos \frac{\theta}{2} | S_z = \frac{1}{2} \rangle_i + \sin \frac{\theta}{2} e^{i\phi} | S_z = -\frac{1}{2} \rangle_i \right)$,

---

[5]The Mermin-Wagner theorem forbids such a symmetry-breaking order for $d = 1$.

which has a $U(1)$ symmetry generated by $\hat{N} = \sum_i \left( \cos\theta \hat{S}_i^z + \sin\theta \cos\phi \hat{S}_i^x + \sin\theta \sin\phi \hat{S}_i^y \right)$. Therefore, $SO(3)$ can never be broken down to $\mathbb{Z}_2$ if $S = 1/2$ and if the ground state can be represented by a local product state. We therefore see that for $S = 1/2$ the $G_0 \to G$ symmetry-breaking order must be entanglement-enabled if it can be realized.

## 5.2 EESBO for $S = \frac{1}{2}$

A representative wavefunction of this EESBO can be taken as a stack of spin-1/2 chains (see Fig. 4), where each chain is described by a matrix-product state

$$|\psi\rangle = \sum_{\{s_j\}} \text{tr}(A^{[s_1]} \dots A^{[s_L]}) |s_1 \dots s_L\rangle , \tag{3}$$

with matrices

$$A^{[s_z=\frac{1}{2}]} = \begin{pmatrix} 1+a & 0 \\ 0 & 1-a \end{pmatrix}, \qquad A^{[s_z=-\frac{1}{2}]} = \begin{pmatrix} 0 & b \\ c & 0 \end{pmatrix}. \tag{4}$$

Each chain has a translation symmetry along itself, and different chains are arranged so that the entire system has the $\mathbb{Z}^d$ translation symmetry. If $a = b = c = 0$, this state is a product state (and therefore not an EESBO) with all spins pointing in the $z$-direction, and it has the aforementioned remaining unbroken $U(1)$ and translation symmetries. To make the wavefunction an EESBO with the unbroken $\mathbb{Z}_2$ and translation symmetry, we consider nonzero $a$, $b$ and $c$. In Appendix C, we show that for generic nonzero real-valued $a$, $b$ and $c$, the remaining symmetry of this system is $G = \mathbb{Z}_2 \times \mathbb{Z}^d$ and that the state is short-range entangled.

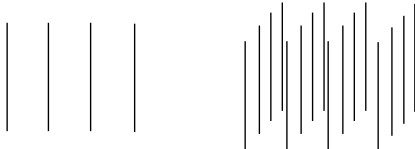

Figure 4: Stacks of spin-1/2 chains that realize the $G_0 = SO(3) \times \mathbb{Z}^d \to G = \mathbb{Z}_2 \times \mathbb{Z}^d$ EESBOs in 2 (left) and 3 (right) dimensions, where each chain is described by the matrix-product state in Eq. (4).

A remarkable aspect of this wavefunction is that it can be viewed as a $G$-SPT beyond the conventional group-cohomology-based classification in Refs. [22, 23, 28] and in some sense even beyond the more refined classification in Refs. [18, 24], which is possible because the $\mathbb{Z}_2$ symmetry acts on states as a $\mathbb{Z}_4$ group. Furthermore, in these symmetry-breaking orders, for all $S \geq 1/2$, there are generically two gapless Goldstone modes, where one of them has a linear dispersion and the other has a quadratic dispersion. We present these analyses in Appendix C.

This example highlights some intriguing points. First, in the previous examples, after fixing $G$, $G_0$ is immaterial in determining whether the $G_0 \to G$ symmetry-breaking order is entanglement-enabled. However, the nature of $G_0$ is important here. For instance, if the $SO(3)$ spin rotation symmetry is replaced by $O(2)$, $G_0 = O(2) \times \mathbb{Z}^d$ still has $G = \mathbb{Z}_2 \times \mathbb{Z}^d$ as its subgroup, but the $G_0 \to G$ symmetry-breaking order is "classical" for all $S$.[6] Second, it is interesting to see that fixing $G_0$ but enlarging $G = \mathbb{Z}_2 \times \mathbb{Z}^d$ to $G = U(1) \times \mathbb{Z}^d$ makes the would-be $G_0 \to G$ EESBO for $S = 1/2$ "classical". In addition, in the previous examples, after *explicitly* breaking $G_0$ while preserving $G$, the EESBOs cannot smoothly evolve into local product states.

---

[6]For example, the $O(2) = SO(2) \rtimes \mathbb{Z}_2$ can be taken to include the $SO(2)$ spin rotation around $S_x$ and the $\mathbb{Z}_2$ spin rotation with respect to $S_z$ by $\pi$. In the symmetry-breaking order under consideration, the $SO(2)$ is completely broken while the $\mathbb{Z}_2$ is preserved. A representative state can be taken as $\bigotimes_i |S_z = \frac{1}{2}\rangle_i$, which is an ultralocal product state.

However, here the EESBO *can* be evolved into a local product state by tuning $b$ and $c$ to zero (though at which point it has an enlarged $U(1) \times \mathbb{Z}^d$ symmetry). Third, this example shows that whether a symmetry-breaking order is entanglement-enabled can depend on the precise representation of DOF under the on-site symmetry, rather than only the equivalence class of projective representations.

### 5.3 Additional examples

Here we present two more examples of EESBOs with spontaneously broken continuous symmetries (but these by no means exhaust all such EESBOs).

The first example is a $d$-dimensional ($d > 1$) spin-1 lattice system with one site per unit cell which may be realized in some generalizations of the bilinear-biquadratic model (see, e.g., Ref. [29] for a discussion of the bilinear-biquadratic model). Suppose its Hamiltonian has $PSU(3)$ and translation symmetries, i.e., $G_0 = PSU(3) \times \mathbb{Z}^d$, where each site of the lattice hosts a spin-1 moment that transforms in the fundamental representation of $SU(3)$, which is a projective representation of $PSU(3)$. Consider the case $G = SO(3) \times \mathbb{Z}^d$, i.e., the translation is intact while $PSU(3)$ is spontaneously broken to $SO(3)$. Here the translation symmetry ensures that all local product states must be ultralocal, because if any two sites were entangled, translation symmetry requires that extensively many sites are entangled. But ultralocal product states are clearly not invariant under the remaining on-site $SO(3)$ symmetry. So this symmetry-breaking order is entanglement-enabled. Its representative wave function can be simply taken as a stack of many copies of the Affleck-Kennedy-Lieb-Tasaki (AKLT) chains arranged to have the $\mathbb{Z}^d$ translation symmetry [30]. If desired, one can also construct a cat state that preserves the full $G_0$ symmetry by summing over the symmetry-breaking orbits. Because each AKLT is short-range entangled, the entire stack is also short-range entangled. In other words, there is no LSM constraint associated with the remaining symmetry $G$. Also, this representative wavefunction shows that this EESBO is a within-group-cohomology state under the remaining symmetry $G$.

The second example is a honeycomb lattice spin-1/2 system with $G_0 = SO(3) \times \mathbb{Z}_2^T \times p6m$, where the spins live on the sites of a honeycomb lattice with a $p6m$ lattice symmetry, and they carry spin-1/2 under the $SO(3)$ spin rotation symmetry and transform as a Kramers doublet under the $\mathbb{Z}_2^T$ time reversal symmetry. Such a system is free of LSM constraints [31]. Now consider a spin-nematic order where the remaining symmetry $G = \mathbb{Z}_2 \times \mathbb{Z}_2 \times \mathbb{Z}_2^T \times p6m$, i.e., the lattice symmetry and time reversal symmetry are intact, while the $SO(3)$ symmetry is broken to $\mathbb{Z}_2 \times \mathbb{Z}_2$ generated by $\pi$ spin rotations around three orthogonal axes. It is straightforward to check that the lattice symmetry again requires a local product state to be ultralocal, which must then break the remaining $\mathbb{Z}_2 \times \mathbb{Z}_2 \times \mathbb{Z}_2^T$ on-site symmetry, because each spin transforms in its projective representation. So such a spin-nematic order is entanglement-enabled. To obtain the wave function of this state, one can start from the $G_0$-symmetric wavefunction in Ref. [31] and perturb it so that $G_0$ is broken to $G$. As a spin-nematic order, there are gapless Goldstone modes carrying spin-2. Furthermore, Ref. [32] predicted that the vicinity of this EESBO may realize an exotic quantum spin liquid that cannot be described by the usual parton approach.

## 6 Discussion

We have discussed various examples of EESBOs, which are intrinsically quantum symmetry-breaking orders that elude the standard "classical" descriptions. While one mechanism of EESBO is due to the LSM constraints of the remaining symmetry, we present a criterion for EESBO in the cases without LSM constraints. Our work paves the way to discovering more

interesting aspects and insightful results about EESBOs. We finish this paper by discussing some future directions.

For our infinite family of one dimensional EESBOs, it is interesting to generalize them to higher dimensions. Since these EESBOs have symmetry-protected gapless boundaries, it is natural to expect the transitions out of these phases to be gapless SPT states [33–35], which warrant a further study. For the EESBOs with spontaneously broken continuous symmetries, finding their parent Hamiltonians is another important problem.

It might also be rewarding to connect the intrinsic quantumness of EESBOs to the complexity of (path-integral) quantum Monte Carlo calculations from the perspective of quantum-classical mapping. The Hamiltonian Eq. (1) is not stoquastic in either the $\mu^z/\nu^z$-diagonal basis or the $\mu^x/\nu^x$-diagonal basis, though the ground state wavefunctions are nonnegative in both basis. What would a corresponding "classical statistical mechanics model" look like for EESBO, if such a sign problem of its Hamiltonian is curable? Could the intrinsic quantumness of EES-BOs suggest that it is not curable by some (symmetry-preserving) local basis transformation, signifying the sign-problem being symmetry protected or even intrinsic [36–39]?

Although much of the universal low-energy physics of EESBOs can be captured by the usual effective field theory approach, it may require new methods to understand other important properties of an EESBO due to the absence of a "classical" mean-field description. For example, in EESBOs with spontaneously broken continuous symmetries, it is desirable to understand the band structures of their Goldstone modes. The conventional method is to analyze the effective theory of these Goldstone modes based on a "classical" mean field, which is unavailable for EESBOs. To proceed, one may need techniques like the parton approach and/or tensor networks. It is also interesting to understand the entanglement-enabled nature of these symmetry-breaking orders from a field theoretic perspective, similar to the description of LSM constraints using topological field theories [24, 28, 32].

A more formal challenge is to classify all EESBOs. From our examples, we see that EESBOs often arise when the local DOF carry specific representations under the symmetry. This is analogous to the physics of obstructed insulators, fragile topology and generalized LSM constraints studied in the context of *symmetric* states [18, 40–44], because in all these cases certain many-body entanglement pattern is incompatible with the properties of local DOF. It is interesting to see if this analogy can be made sharper and useful. For example, one can ask how the EESBO ground states with spontaneously broken continuous symmetries constrain the possible band topology of the Goldstone modes. To this end, anomaly-based methods developed in Refs. [32, 45] may be helpful.

Finally, it is important to identify more models and experimental systems exhibiting EES-BOs. For example, it might be worth checking if some of the enigmatic "hidden orders" turn out to be EESBOs. It is also interesting to see if various quantum simulators can realize EES-BOs. Moreover, it is worth exploring whether the corresponding symmetry defects, whose properties are not well understood so far, have potential technological applications.

## Acknowledgements

We thank Xie Chen, Tyler Ellison, Yin-Chen He, Timothy Hsieh, Olexei Motrunich, Chao-Ming Jian, Shenghan Jiang, Yang Qi, Chong Wang, Cenke Xu and Yi-Zhuang You for valuable discussions and feedback.

**Funding information**    Research at Perimeter Institute is supported in part by the Government of Canada through the Department of Innovation, Science and Economic Development Canada and by the Province of Ontario through the Ministry of Colleges and Universities. C.-J. Lin acknowledges the support from the National Science Foundation (QLCI grant OMA-2120757).

## A  Examples of EESBOs from the previous literature

In this appendix, we review some examples of EESBOs from the previous literature [2–4]. In particular, we will apply our diagnostics to verify that these symmetry-breaking orders are indeed entanglement-enabled.

We start with the one dimensional example in Ref. [2], which has very similar flavor as the examples presented in Sec. 2. The relevant symmetry $G_0$ in this example includes a $PSU(4)$ internal symmetry and a $p1m$ lattice symmetry. Each site hosts a 6-dimensional Hilbert space, and the states at each site transform as a vector under the $SO(6)$ symmetry (while the operators transform in the fundamental linear representation under $PSU(4)$), so these states are in the projective representation of $PSU(4)$. Furthermore, the $p1m$ symmetry moves the locations of the degrees of freedom in the same way as the example in Sec. 4. In addition, the ground state breaks bond-centered reflection symmetry, just like the example in Sec. 4. Combining all these observations, an argument almost identical to the one in Sec. 4 shows that the symmetry-breaking order in Ref. [2] is entanglement-enabled.

Next, we discuss the example in Ref. [3], which proposed a spin nematic phase in a model on a square lattice spin-1/2 system with $G_0 = SO(3) \times \mathbb{Z}_2^T \times p4m$. The actions of the symmetry on the degrees of freedom is the same as in the familiar square lattice Heisenberg model. The complete remaining symmetry $G$ is not explicitly discussed in Ref. [3], but it appears that $G = \mathbb{Z}_2 \times \mathbb{Z}_2^T \times pmm$, i.e., the $SO(3)$ spin rotational symmetry is broken to $\mathbb{Z}_2$ and the $p4m$ lattice symmetry is broken to $pmm$, while the time reversal symmetry is preserved. If this is indeed the unbroken symmetry in this order, then there is still a nontrivial LSM constraint in the system that enforces all $G$-symmetric ground states to be long-range entangled. That is, this spin nematic order is a long-range EESBO.

However, various later papers question whether the model in Ref. [3] actually yields this spin nematic phase (see Ref. [4] for a recent study). The new consensus is that upon adding a mangetic field that explicitly breaks $G_0 = SO(3) \times \mathbb{Z}_2^T \times p4m$ in Ref. [3] to $G_0 = U(1) \times p4m$, the ground state spontaneously breaks this new $G_0$ to $G = \mathbb{Z}_2 \times pmm$. This symmetry-breaking order is also entanglement-enabled, and it has a similar flavor as the example in Sec. 5. This is because there is also no LSM constraint from the remaining symmetry, and the lattice symmetry also forces all local product states to be ultralocal. Then an almost identical argument as that in Sec. 5 shows that this is a short-range EESBO.

In passing, we note that the entanglement-enabled nature of the spin nematic state proposed in Ref. [3] is independent of whether the model therein realizes this order. That is, as long as this order is realized in any model with the same symmetry setting as in Ref. [3], then it is entanglement-enabled, since our argument only relies on the symmetry breaking pattern.

## B  More on the one dimensional Hamiltonians

In this appendix, we show several properties of the 1d Hamiltonian Eq. (1) in the main text. We will show that (i) the only ground states of these Hamiltonians are the ones given in the main text, (ii) all these Hamiltonians are gapped, (iii) the ground states are nontrivial symmetry-protected topological (SPT) states under the remaining $\mathbb{Z}_n^x \times \mathbb{Z}_n^z$ symmetry, and (iv) under the remaining $\mathbb{Z}_n^x \times \mathbb{Z}_n^z \times \sigma$ symmetry they are beyond the conventional group-cohomology-based classification developed in Refs. [22, 23]. Note that the first three properties apply to the Hamiltonians defined for any $n \geqslant 2$, while the last one applies to the case with $n \geqslant 3$. In the main text, we have shown that the relevant $G_0 \to G$ symmetry breaking orders are entanglement-enabled when $n \geqslant 3$. For $n = 2$, the $G_0 \to G$ symmetry breaking order can be realized by local product states, i.e., $\prod_j(|00\rangle_j + (-1)^j|11\rangle_j)$ with $j$ the site index, so it is not

entanglement-enabled.

Here we repeat the Hamiltonian for reader's convenience:

$$H_L = \sum_{j=1}^{L-1} (I - P_{j,j+1}) + r(I - P_{L,1}) \,, \tag{5}$$

where $P_{j,j+1}$ is the projector onto the linear space $\mathcal{P}_{j,j+1} = \mathrm{span}\{|D_{a,b}^{(\alpha)}\rangle, |D_{a,b}^{(\beta)}\rangle; a, b = 1 \ldots n\}$ and

$$
\begin{aligned}
|D_{a,b}^{(\alpha)}\rangle &= \sum_{d=1}^{n} |\alpha_j = d, \beta_j = a, \alpha_{j+1} = d, \beta_{j+1} = b\rangle \,, \\
|D_{a,b}^{(\beta)}\rangle &= \sum_{d=1}^{n} |\alpha_j = a, \beta_j = d, \alpha_{j+1} = b, \beta_{j+1} = d\rangle \,,
\end{aligned}
$$

are the maximum entangled states on $\alpha$ or $\beta$ degrees of freedom. We may refer to the space spanned by these states the "dimer" subspace of the $\alpha$ or $\beta$ degrees of freedom, respectively. In particular, we will consider the Hamiltonian with a open boundary condition ($r = 0$) or a periodic boundary condition ($r = 1$).

It is sometimes useful to express the projector in terms of the operators [18]. Consider the projectors projecting the states into the dimer subspace on $\alpha$ or $\beta$ degrees of freedom $P_{j,j+1}^{(\alpha/\beta)}$, we have $P_{j,j+1}^{(\alpha/\beta)} = P_{(\alpha/\beta)}^{x} P_{(\alpha/\beta)}^{z}$, where

$$P_{(\alpha)}^{x} = \frac{1}{n} \sum_{d=0}^{n-1} (\mu_j^x \mu_{j+1}^x)^d \quad \text{and} \quad P_{(\alpha)}^{z} = \frac{1}{n} \sum_{d=0}^{n-1} (\mu_j^z)^d (\mu_{j+1}^z)^{-d} \,, \tag{6}$$

while

$$P_{(\beta)}^{x} = \frac{1}{n} \sum_{d=0}^{n-1} (\nu_j^x \nu_{j+1}^x)^d \quad \text{and} \quad P_{(\beta)}^{z} = \frac{1}{n} \sum_{d=0}^{n-1} (\nu_j^z)^d (\nu_{j+1}^z)^{-d} \,. \tag{7}$$

We then have $P_{j,j+1} = P_{j,j+1}^{(\alpha)} + P_{j,j+1}^{(\beta)} - P_{j,j+1}^{(\alpha)} P_{j,j+1}^{(\beta)}$.

## B.1 Uniqueness of the ground states

First we show the uniqueness of the ground states. Consider the following states depicted in Fig. 2,

$$
\begin{aligned}
|\psi_{a,b}^{(A)}\rangle &= \sum_{\{\alpha_j, \beta_j\}} \psi_{\{\alpha_j \beta_j\}}^{(A)}(a,b) |\alpha_1 \beta_1 \ldots \alpha_L \beta_L\rangle \,, \\
|\psi_{a,b}^{(B)}\rangle &= \sum_{\{\alpha_j, \beta_j\}} \psi_{\{\alpha_j \beta_j\}}^{(B)}(a,b) |\alpha_1 \beta_1 \ldots \alpha_L \beta_L\rangle \,, \quad a, b = 1 \ldots n \,,
\end{aligned} \tag{8}
$$

where

$$
\begin{aligned}
\psi_{\{\alpha_j, \beta_j\}}^{(A)}(a,b) &= \delta_{\alpha_1 a} \left( \prod_{j=1}^{L/2-1} \delta_{\beta_{2j-1} \beta_{2j}} \delta_{\alpha_{2j} \alpha_{2j+1}} \right) \delta_{\beta_{L-1} \beta_L} \delta_{\alpha_L b} \,, \\
\psi_{\{\alpha_j, \beta_j\}}^{(B)}(a,b) &= \delta_{\beta_1 a} \left( \prod_{j=1}^{L/2-1} \delta_{\alpha_{2j-1} \alpha_{2j}} \delta_{\beta_{2j} \beta_{2j+1}} \right) \delta_{\alpha_{L-1} \alpha_L} \delta_{\beta_L b} \,, \quad \text{if } L \text{ is even}, \tag{9}
\end{aligned}
$$

and

$$\psi^{(A)}_{\{\alpha_j,\beta_j\}}(a,b) = \delta_{\alpha_1 a}\left(\prod_{j=1}^{(L-1)/2}\delta_{\beta_{2j-1}\beta_{2j}}\delta_{\alpha_{2j}\alpha_{2j+1}}\right)\delta_{\beta_L b},$$

$$\psi^{(B)}_{\{\alpha_j,\beta_j\}}(a,b) = \delta_{\beta_1 a}\left(\prod_{j=1}^{(L-1)/2}\delta_{\alpha_{2j-1}\alpha_{2j}}\delta_{\beta_{2j}\beta_{2j+1}}\right)\delta_{\alpha_L b}, \quad \text{if } L \text{ is odd}. \tag{10}$$

Since the Hamiltonian is of the form of a sum of projectors, the eigenvalues of $H_L$ is bounded below by 0. For the open boundary condition, it is easy to verify $H_L|\psi^{(A)}_{a,b}\rangle = 0$ and $H_L|\psi^{(B)}_{a,b}\rangle = 0$ for all $a, b = 1\ldots n$. We therefore would like to show that $\mathrm{Ker}(H_L) = \mathcal{G}_L$, where the span of the states is $\mathcal{G}_L \equiv \mathrm{span}\{|\psi^{(A)}_{a,b}\rangle, |\psi^{(B)}_{a,b}\rangle, a, b = 1\ldots n\}$. Note that these $2n^2$ states are linearly independent (but not orthogonal) when $L \geqslant 3$, so $\dim(\mathcal{G}_L) = 2n^2$ when $L \geqslant 3$. If $L = 2$, $\dim(\mathcal{G}_L) = 2n^2 - 1$.

To show $\mathrm{Ker}(H_L) = \mathcal{G}_L$ for any $L \geqslant 2$, we use mathematical induction. Assume that for a 1d lattice with $L$ sites (onsite Hilbert space dimension $n^2$) and the corresponding Hilbert space $\mathcal{H}_L$, the ground state space $\mathrm{Ker}(H_L) = \mathcal{G}_L \subset \mathcal{H}_L$. Now we would like to add one more site and to find $\mathrm{Ker}(H_{L+1}) = \mathrm{Ker}(H_L) \cap \mathrm{Ker}(I - P_{L,L+1})$, where both $\mathrm{Ker}(H_L)$ and $\mathrm{Ker}(I - P_{L,L+1})$ should be understood as a linear subspace in $\mathcal{H}_{L+1}$. (This means $\dim(\mathrm{Ker}(H_L)) = 2n^2 \times n^2$ as a linear subspace of $\mathcal{H}_{L+1}$.)

To find the intersection of the two linear subspaces, assume $|\psi\rangle \in \mathrm{Ker}(H_L) \subset \mathcal{H}_{L+1}$. Let us first consider the case where $L$ is odd. We can write

$$|\psi\rangle = \sum_{\{\alpha_j\beta_j\}}\left(A_{\alpha_1\beta_L\alpha_{L+1}\beta_{L+1}}\chi^{(A)}_{\beta_1\alpha_2\ldots\alpha_L} + B_{\beta_1\alpha_L\alpha_{L+1}\beta_{L+1}}\chi^{(B)}_{\alpha_1\alpha_2\beta_2\ldots\beta_{L-1}\beta_L}\right)|\alpha_1\beta_1\ldots\alpha_{L+1}\beta_{L+1}\rangle, \tag{11}$$

where

$$\chi^{(A)}_{\beta_1\alpha_2\ldots\alpha_L} = \prod_{j=1}^{(L-1)/2}\delta_{\beta_{2j-1}\beta_{2j}}\delta_{\alpha_{2j}\alpha_{2j+1}} \quad \text{and} \quad \chi^{(B)}_{\alpha_1\alpha_2\beta_2\ldots\beta_{L-1}\beta_L} = \prod_{j=1}^{(L-1)/2}\delta_{\alpha_{2j-1}\alpha_{2j}}\delta_{\beta_{2j}\beta_{2j+1}},$$

(the dimer part of the wavefunction), and there are $2n^2 \times n^2$ coefficients $A_{\alpha_1\beta_L\alpha_{L+1}\beta_{L+1}}$ and $B_{\beta_1\alpha_L\alpha_{L+1}\beta_{L+1}}$. Requiring $|\psi\rangle \in \mathrm{Ker}(I - P_{L,L+1})$ as well, we can also express

$$|\psi\rangle = \sum_{\{\alpha_j\beta_j\}}\left(X_{\alpha_1\beta_1\ldots\alpha_L\alpha_{L+1}}\delta_{\beta_L\beta_{L+1}} + Y_{\alpha_1\beta_1\ldots\beta_L\beta_{L+1}}\delta_{\alpha_L\alpha_{L+1}}\right)|\alpha_1\beta_1\ldots\alpha_{L+1}\beta_{L+1}\rangle, \tag{12}$$

where $X_{\alpha_1\beta_1\ldots\alpha_L\alpha_{L+1}}$ and $Y_{\alpha_1\beta_1\ldots\beta_L\beta_{L+1}}$ are the coefficients of the linear combination of the vectors in $\mathrm{Ker}(I - P_{L,L+1})$.

Equating the above two equations Eqs. (11) and (12), we obtain $n^{2(L+1)}$ linear equations $(\alpha_j, \beta_j = 1\ldots n)$

$$\begin{aligned} &A_{\alpha_1\beta_L\alpha_{L+1}\beta_{L+1}}\chi^{(A)}_{\beta_1\alpha_2\ldots\alpha_L} + B_{\beta_1\alpha_L\alpha_{L+1}\beta_{L+1}}\chi^{(B)}_{\alpha_1\alpha_2\beta_2\ldots\beta_{L-1}\beta_L}\\ =\ &X_{\alpha_1\beta_1\ldots\alpha_L\alpha_{L+1}}\delta_{\beta_L\beta_{L+1}} + Y_{\alpha_1\beta_1\ldots\beta_L\beta_{L+1}}\delta_{\alpha_L\alpha_{L+1}}, \end{aligned}$$

which pose constraints on $A_{\alpha_1\beta_L\alpha_{L+1}\beta_{L+1}}$ and $B_{\beta_1\alpha_L\alpha_{L+1}\beta_{L+1}}$ (or $X_{\alpha_1\beta_1\ldots\alpha_L\alpha_{L+1}}$ and $Y_{\alpha_1\beta_1\ldots\beta_L\beta_{L+1}}$ conversely). For convenience, we will use the notation $\bar{\alpha}_j$ to denote the numbers in the set $\{\bar{\alpha}_j = 1\ldots n, \bar{\alpha}_j \neq \alpha_j\}$ and similarly for $\bar{\beta}_j$.

First, considering the subset of the equations where $\beta_{2j} = \beta_{2j-1}$, $\alpha_{2j+1} = \alpha_{2j}$ for $j = 1\ldots(L-1)/2$ (so that $\chi^{(A)} = 1$), $\beta_L \neq \beta_{L+1}$, $\alpha_L \neq \alpha_{L+1}$ (so that the right-hand side

is zero) and $\beta_{L-1} \neq \beta_L$ (so taht $\chi^{(B)} = 0$), we have $A_{\alpha_1 \beta_L \alpha_{L+1} \bar{\beta}_L} = 0$. Similarly, the subset of the equations where $\alpha_{2j} = \alpha_{2j-1}$, $\beta_{2j+1} = \beta_{2j}$ for $j = 1 \ldots (L-1)/2$ (so that $\chi^{(B)} = 1$), $\beta_L \neq \beta_{L+1}$, $\alpha_L \neq \alpha_{L+1}$ (so that the right-hand side is zero) and $\alpha_{L-1} \neq \alpha_L$ (so that $\chi^{(A)} = 0$) give us

$$B_{\beta_1 \alpha_L \bar{\alpha}_L \beta_{L+1}} = 0 \ .$$

Next, we consider again the equations where $\beta_{2j} = \beta_{2j-1}$, $\alpha_{2j+1} = \alpha_{2j}$ for $j = 1 \ldots (L-1)/2$ (so that $\chi^{(A)} = 1$), but now with $\beta_L = \beta_{L+1}$ and $\alpha_L \neq \alpha_{L+1}$ (so that $B = 0$), giving us

$$A_{\alpha_1 \beta_L \alpha_{L+1} \beta_L} = X_{\alpha_1 \beta_1 \ldots \bar{\alpha}_{L+1} \alpha_{L+1}} \ .$$

Note that on the right-hand side, it is independent of $\beta_L$ (but depending on $\alpha_1$ and $\alpha_{L+1}$). We therefore conclude that $A_{\alpha_1 \beta_L \alpha_{L+1} \beta_{L+1}} = a_{\alpha_1 \alpha_{L+1}} \delta_{\beta_L \beta_{L+1}}$. Similarly, we consider the equations where $\alpha_{2j} = \alpha_{2j-1}$, $\beta_{2j+1} = \beta_{2j}$ for $j = 1 \ldots (L-1)/2$ (so that $\chi^{(B)} = 1$), but now with $\beta_L \neq \beta_{L+1}$ (so that $A = 0$) and $\alpha_L = \alpha_{L+1}$. We have

$$B_{\beta_1 \alpha_L \alpha_L \beta_{L+1}} = Y_{\alpha_1 \beta_1 \ldots \bar{\beta}_{L+1} \beta_{L+1}} \ .$$

Again, since it is independent of $\alpha_L$ on the right-hand side, we conclude that

$$B_{\beta_1 \alpha_L \alpha_{L+1} \beta_{L+1}} = b_{\beta_1 \beta_{L+1}} \delta_{\alpha_L \alpha_{L+1}} \ .$$

This shows that $\mathcal{G}_{L+1} \supseteq \mathrm{Ker}(H_{L+1})$. Since we already know $\mathcal{G}_{L+1} \subseteq \mathrm{Ker}(H_{L+1})$, we indeed have $\mathcal{G}_{L+1} = \mathrm{Ker}(H_{L+1})$.

If $L$ is even, a similar argument can be made with some modification. In particular, the set of linear equations constraining the coefficients $A$, $B$, $X$ and $Y$ are now

$$A_{\alpha_1 \alpha_L \alpha_{L+1} \beta_{L+1}} \chi^{(A)}_{\beta_1 \alpha_2 \ldots \beta_{L-1} \beta_L} + B_{\beta_1 \beta_L \alpha_{L+1} \beta_{L+1}} \chi^{(B)}_{\alpha_1 \alpha_2 \beta_2 \ldots \alpha_L \beta_L} = X_{\alpha_1 \beta_1 \ldots \alpha_L \alpha_{L+1}} \delta_{\beta_L \beta_{L+1}} + Y_{\alpha_1 \beta_1 \ldots \beta_L \beta_{L+1}} \delta_{\alpha_L \alpha_{L+1}} \ ,$$

where

$$\chi^{(A)}_{\beta_1 \alpha_2 \ldots \beta_{L-1} \beta_L} = \prod_{j=1}^{L/2-1} \delta_{\beta_{2j-1} \beta_{2j}} \delta_{\alpha_{2j} \alpha_{2j+1}} \delta_{\beta_{L-1} \beta_L} \quad \text{and} \quad \chi^{(B)}_{\alpha_1 \alpha_2 \beta_2 \ldots \alpha_L \beta_L} = \prod_{j=1}^{L/2-1} \delta_{\alpha_{2j-1} \alpha_{2j}} \delta_{\beta_{2j} \beta_{2j+1}} \delta_{\alpha_{L-1} \alpha_L} \ .$$

Again, considering the subset of equations where $\beta_{2j-1} = \beta_{2j}$ for $j = 1 \ldots L/2$, $\alpha_{2j} = \alpha_{2j+1}$ for $j = 1 \ldots L/2-1$, $\beta_L \neq \beta_{L+1}$, $\alpha_L \neq \alpha_{L+1}$ and $\alpha_{L-1} \neq \alpha_L$, we have $A_{\alpha_1 \alpha_L \bar{\alpha}_L \beta_{L+1}} = 0$; while considering the subset of equations where $\alpha_{2j-1} = \alpha_{2j}$ for $j = 1 \ldots L/2$, $\beta_{2j} = \beta_{2j+1}$ for $j = 1 \ldots L/2-1$, $\alpha_L \neq \alpha_{L+1}$, $\beta_L \neq \beta_{L+1}$ and $\beta_{L-1} \neq \beta_L$, we have $B_{\beta_1 \beta_L \alpha_L \bar{\beta}_{L+1}} = 0$.

Now the set of equations where $\beta_{2j-1} = \beta_{2j}$ for $j = 1 \ldots L/2$, $\alpha_{2j} = \alpha_{2j+1}$ for $j = 1 \ldots L/2-1$, $\beta_L \neq \beta_{L+1}$, $\alpha_L = \alpha_{L+1}$ give us $A_{\alpha_1 \alpha_L \alpha_L \beta_{L+1}} = Y_{\alpha_1 \ldots \bar{\beta}_{L+1} \beta_{L+1}}$, resulting in

$$A_{\alpha_1 \alpha_L \alpha_{L+1} \beta_{L+1}} = a_{\alpha_1 \beta_{L+1}} \delta_{\alpha_L \alpha_{L+1}} \ .$$

Similarly, the set of equations where $\alpha_{2j-1} = \alpha_{2j}$ for $j = 1 \ldots L/2$, $\beta_{2j} = \beta_{2j+1}$ for $j = 1 \ldots L/2-1$, $\beta_L = \beta_{L+1}$, $\alpha_L \neq \alpha_{L+1}$ give us $B_{\beta_1 \beta_L \alpha_{L+1} \beta_L} = X_{\alpha_1 \ldots \bar{\alpha}_{L+1} \alpha_{L+1}}$, resulting in

$$B_{\beta_1 \beta_L \alpha_{L+1} \beta_{L+1}} = b_{\beta_1 \alpha_{L+1}} \delta_{\beta_L \beta_{L+1}} \ .$$

We therefore conclude $\mathcal{G}_{L+1} = \mathrm{Ker}(H_{L+1})$ if $L$ is even. The mathematical induction is therefore establish since $\mathcal{G}_2 = \mathrm{Ker}(H_2) = \mathrm{Ker}(I - P_{1,2})$.

Finally, we show that if $L \geq 4$ and even, and $r = 1$ (periodic boundary condition), the ground state space is spanned by the states depicted in the main text. We would like to find $\mathrm{Ker}[H_L(r = 1)] = \mathrm{Ker}[H_L(r = 0)] \cap \mathrm{Ker}(I - P_{L,1}) \subset \mathcal{H}_L$.

By expressing $|\psi\rangle$ as

$$
\begin{aligned}
|\psi\rangle &= \sum_{\{\alpha_j\beta_j\}} \left( A_{\alpha_1\alpha_L} \chi^{(A)}_{\beta_1\alpha_2...\alpha_L} + B_{\beta_1\beta_L} \chi^{(B)}_{\alpha_1\alpha_2\beta_2...\beta_{L-1}\beta_L} \right) |\alpha_1\beta_1...\alpha_{L+1}\beta_{L+1}\rangle \\
&= \sum_{\{\alpha_j\beta_j\}} \left( X_{\alpha_1\alpha_2\beta_2...\alpha_L} \delta_{\beta_L\beta_1} + Y_{\beta_1\alpha_2...\beta_L} \delta_{\alpha_L\alpha_1} \right) |\alpha_1\beta_1...\alpha_{L+1}\beta_{L+1}\rangle ,
\end{aligned}
\tag{13}
$$

where

$$
\chi^{(A)}_{\beta_1\alpha_2...\beta_{L-1}\beta_L} = \prod_{j=1}^{L/2-1} \delta_{\beta_{2j-1}\beta_{2j}} \delta_{\alpha_{2j}\alpha_{2j+1}} \delta_{\beta_{L-1}\beta_L} \quad \text{and} \quad \chi^{(B)}_{\alpha_1\alpha_2\beta_2...\alpha_L\beta_L} = \prod_{j=1}^{L/2-1} \delta_{\alpha_{2j-1}\alpha_{2j}} \delta_{\beta_{2j}\beta_{2j+1}} \delta_{\alpha_{L-1}\alpha_L} ,
$$

we have the linear equations

$$
A_{\alpha_1\alpha_L} \chi^{(A)}_{\beta_1\alpha_2...\alpha_L} + B_{\beta_1\beta_L} \chi^{(B)}_{\alpha_1\alpha_2\beta_2...\beta_{L-1}\beta_L} = X_{\alpha_1\alpha_2\beta_2...\alpha_L} \delta_{\beta_L\beta_1} + Y_{\beta_1\alpha_2...\beta_L} \delta_{\alpha_L\alpha_1} .
\tag{14}
$$

The equations such that $\chi^{(A)}_{\beta_1\alpha_2...\beta_{L-1}\beta_L} = 1$, $\alpha_{L-1} \neq \alpha_L$ and $\beta_1 \neq \beta_L$ give us

$$
A_{\alpha_1\alpha_L} = Y_{...} \delta_{\alpha_1\alpha_L} \equiv a\delta_{\alpha_1\alpha_L} ,
$$

while the equations such that $\chi^{(B)}_{\alpha_1\alpha_2\beta_2...\beta_{L-1}\beta_L} = 1$, $\beta_{L-1} \neq \beta_L$ and $\alpha_1 \neq \alpha_L$ give us

$$
B_{\beta_1\beta_L} = X_{...} \delta_{\beta_1\beta_L} \equiv b\delta_{\beta_1\beta_L} .
$$

## B.2 Existence of the spectral gap in the thermodynamic limit

To show the existence of the spectral gap in the thermodynamic limit, we will use Theorem 2 in Ref. [46]. Our ground states in a open chain are indeed a type of the *generalized valence bond solid* (GVBS) states defined in Ref. [46]. Consider an interval $[M, N]$ in an infinite 1d lattice, we can define our Hamiltonian in this subregion $H_{[M,N]} = \sum_{j=M}^{N-1} h_j$, where $h_j = (1 - P_{j,j+1})$. Now in the previous section, we have shown that the ground state subspace is indeed the GVBS states (condition $\mathcal{F}_\omega = \mathcal{F}_h$ in Theorem 2 of Ref. [46]). Due to the projector construction of the Hamiltonian, we also have the condition $\mathrm{Ker}(H)_{[M,N]} = \mathcal{G}_{M,N}$, where $\mathcal{G}_{M,N}$ is the span of the eigenvectors obtained from the reduced density matrix of $\omega$ on the region $[M, N]$. We therefore conclude that our model and ground states satisfy the conditions of Theorem 2 in Ref. [46], which shows the existence of the spectral gap in the thermodynamic limit.

## B.3 Symmetry-protected edge states under open boundary condition

As pointed out in the main text, the ground states of Eq. (5) have dangling edge degrees of freedom, and here we show that they are indeed protected by the symmetry $\mathbb{Z}_n^x \times \mathbb{Z}_n^z$, by showing that these edge degrees of freedom transform in projective representations under $\mathbb{Z}_n^x \times \mathbb{Z}_n^z$.

The ground states under open boundary condition are given by Eq. (8). Recall the symmetry $\mathbb{Z}_n^x$ is implemented by $\prod_j \mu_j^x \nu_j^x$ and $\mathbb{Z}_n^z$ by $\prod_{j\in\text{even}}(\mu_j^z \nu_j^z)\prod_{j\in\text{odd}}(\mu_j^z \nu_j^z)^{-1}$. To determine the effective symmetry actions on the edges, it suffices to examine the action of these symmetries on the "dimer" $\sum_{d=1}^n |dd\rangle$, which is the building block of the bulk. One can easily verify that $\mu_1^x \mu_2^x \sum_{d=1}^n |dd\rangle = \sum_{d=1}^n |dd\rangle$, and similarly for $\nu^x$; similarly, $\mu_1^z(\mu_2^z)^{-1}\sum_{d=1}^n |dd\rangle = \sum_{d=1}^n |dd\rangle$ and likewise for $\nu_1^z(\nu_2^z)^{-1}$. The bulk of the wavefunction is therefore invariant under the symmetry action, and the symmetry action only transforms the edge degrees of freedom. Moreover, the ground states transform among the same $A$-type

$|\psi_{a,b}^{(A)}\rangle$ or same $B$-type of states $|\psi_{a,b}^{(B)}\rangle$. It is then easy to identify the edge symmetry operator. For example, if $L$ is even, within the $A$-type ground state space $\text{span}\{|\psi_{a,b}^{(A)}\rangle\}$,

$$W_L^x = \mu_1^x , \qquad W_R^x = \mu_L^x ,$$
$$W_L^z = (\mu_1^z)^{-1} , \qquad W_R^z = \mu_L^z ;$$

while

$$W_L^x = \nu_1^x , \qquad W_R^x = \nu_L^x ,$$
$$W_L^z = (\nu_1^z)^{-1} , \qquad W_R^z = \nu_L^z ,$$

within the $B$-type ground state space $\text{span}\{|\psi_{a,b}^{(B)}\rangle\}$. The case for an odd $L$ is similar.

It is straightforward to check that the actions of $\mathbb{Z}_n^x$ and $\mathbb{Z}_n^z$ no longer commute on the edge DOF, which means that these edge DOF transform in projective representations under $\mathbb{Z}_n^x \times \mathbb{Z}_n^z$. Projective representations of $\mathbb{Z}_n^x \times \mathbb{Z}_n^z$ are classified by $H^2(\mathbb{Z}_n^x \times \mathbb{Z}_n^z) = \mathbb{Z}_n$, and they can be characterized by an integer $\eta$ that is defined modulo $n$, which is given by $W_{L,R}^x W_{L,R}^z = e^{i\frac{2\pi}{n}\eta_{L,R}} W_{L,R}^z W_{L,R}^x$. Using the above symmetry actions, we see that $\eta_L = -1$ and $\eta_R = 1$, for both the $A$-type and $B$-type ground states. Since the degeneracy between the $A$-type and the $B$-type subspace is protected by the spontaneous breaking of the reflection symmetry $T\sigma$, we conclude that all these edge modes are protected by the unbroken symmetries and spontaneous symmetry breaking.

## B.4 Beyond the group-cohomology-based classification

In the above section, we have shown that the ground states we obtain are nontrivial SPTs under the remaining $\mathbb{Z}_n^x \times \mathbb{Z}_n^z$ symmetry. Since the remaining symmetry in fact contains a $\mathbb{Z}_n^x \times \mathbb{Z}_n^z \times \sigma$ symmetry, we can ask what kind of SPT our ground states are under this symmetry. We will see that our ground states are actually beyond the conventional group-cohomology-based classification developed in Refs. [22, 23].

The argument is a generalization of that in Ref. [18], which only discusses the case with $n = 4$ and has no spontaneous symmetry breaking. The group-cohomology-based classification of $1 + 1$ dimensional $\mathbb{Z}_n^x \times \mathbb{Z}_n^z \times \sigma$ SPTs is

$$H^2(\mathbb{Z}_n^x \times \mathbb{Z}_n^z \times \sigma, U(1)_\sigma) = \begin{cases} \mathbb{Z}_2, & \text{odd } n \\ \mathbb{Z}_2^4, & \text{even } n \end{cases}, \tag{15}$$

where the subscript $\sigma$ in $U(1)$ means that the $U(1)$ phase factor should be complex conjugated when acted by $\sigma$. For odd $n$, the nontrivial SPT is protected only by $\sigma$, which has no protected edge state. Since our states do have protected edge states, they are beyond this classification. For even $n$, the only possible edge states protected by the $\mathbb{Z}_n^x \times \mathbb{Z}_n^z$ symmetry from this classification have $\eta = n/2$. In our case, $\eta = \pm 1$, so our states are also beyond the group-cohomology-based classification if $n > 2$. As noted in Ref. [18], here going beyond the group-cohomology-based classification is possible because the states at each site transform in projective representations of the on-site symmetries. To capture such states, one needs a more refined classification that takes into account the possible projective representations of the states at each site, such as the ones proposed in Refs. [18, 24].

## C The $G_0 = SO(3) \times \mathbb{Z}^d \to G = \mathbb{Z}_2 \times \mathbb{Z}^d$ symmetry-breaking orders

In this apeendix, we present the detailed calculations regarding the EESBO in spin-$S$ lattice system with $G_0 = SO(3) \times \mathbb{Z}^d$ and $G = \mathbb{Z}_2 \times \mathbb{Z}^d$. We will verify that the remaining symmetry of

the wavefunction presented in the main text is indeed $G$. Next, we will show that this wavefunction represents a $G$-symmetric short-range entangled state that is beyond the conventional group-cohomology-based classification of Refs. [22, 23], and in some sense even beyond the more refined classification of Refs. [18, 24]. Finally, we will show that for all $S \geqslant 1/2$, this symmetry-breaking order generically has two gapless Goldstone modes, where one of them has a linear dispersion and the other has a quadratic dispersion.

## C.1  Verification of $G$-symmetry

Here we verify that for the one dimensional matrix product state (MPS) in the main text, represented by the matrices

$$A^{[S_z=\frac{1}{2}]} = \begin{pmatrix} 1+a & 0 \\ 0 & 1-a \end{pmatrix}, \qquad A^{[S_z=-\frac{1}{2}]} = \begin{pmatrix} 0 & b \\ c & 0 \end{pmatrix}, \tag{16}$$

for generic nonzero real-valued $a, b, c$, the state is short-range entangled and that the only subgroup of $SO(3)$ that is still a symmetry of this state is a $\mathbb{Z}_2$ symmetry generated by the $\pi$ rotation with respect to $S_z$. This implies that the stack of chains described in the main text is also a short-range entangled state, whose symmetry is $G = \mathbb{Z}_2 \times \mathbb{Z}^d$.

Showing that this state is short-range entangled amounts to show that this MPS is injective. To this end, we first construct the tensor

$$T_{i,j,k,l} = \sum_{S_z = \pm\frac{1}{2}} A_{ij}^{[S_z]} \left( A_{kl}^{[S_z]} \right)^* , \tag{17}$$

from which we construct the $4 \times 4$ transfer matrix

$$\tilde{T}_{(ik),(jl)} = T_{i,j,k,l} . \tag{18}$$

Here the relation between the indices of the tensor $T$ and those of the transfer matrix $\tilde{T}$ is that $(11) \leftrightarrow 1$, $(12) \leftrightarrow 2$, $(21) \leftrightarrow 3$ and $(22) \leftrightarrow 4$. Using Eqs. (16) and (17), we get

$$\tilde{T} = \begin{pmatrix} (a+1)^2 & 0 & 0 & b^2 \\ 0 & 1-a^2 & bc & 0 \\ 0 & bc & 1-a^2 & 0 \\ c^2 & 0 & 0 & (a-1)^2 \end{pmatrix}, \tag{19}$$

which has eigenvalues $\{1 + a^2 \pm \sqrt{4a^2 + b^2c^2}, 1 - a^2 \pm bc\}$. The largest eigenvalue is non-degenerate for all $a \neq 0$. That is, this MPS is injective and represents a short-range entangled state as long as $a \neq 0$.

Below we show that $\mathbb{Z}_2$ (generated by the $\pi$ rotation with respect to $S_z$) is the only subgroup of the $SO(3)$ spin rotation symmetry that is still a symmetry of this MPS. Our strategy is to first consider an operation whose action on the physical states is given by $U(\theta) = \exp(iS_z\theta)$, where $0 \leqslant \theta < 2\pi$. These are just all spin rotating with respect to $S_z$. We will see that the MPS given by Eq. (16) is invariant only if $\theta = 0, \pi$. This means that the $\mathbb{Z}_2$ generated by the $\pi$ rotation with respect to $S_z$ is indeed a symmetry of this MPS. We then show that this is the only symmetry of that MPS, by showing that $\langle S_x \rangle = \langle S_y \rangle = 0$ while $\langle S_z \rangle \neq 0$ for this MPS.

$U(\theta) = \exp(iS_z\theta)$ is a symmetry of the MPS if and only if [47, 48]

$$U(\theta)_{S_z S_z'} A^{[S_z']} = e^{i\alpha} V A^{[S_z]} V^\dagger , \tag{20}$$

with $\alpha \in \mathbb{R}$ and $V \in SU(2)$. Here $\alpha$ and $V$ can depend on $\theta$. In the eigenbasis of $S_z$, $U(\theta)$ is diagonal, and this equation simply becomes

$$e^{i\frac{\theta}{2}} A^{[S_z=\frac{1}{2}]} = e^{i\alpha} V A^{[S_z=\frac{1}{2}]} V^\dagger, \qquad e^{-i\frac{\theta}{2}} A^{[S_z=-\frac{1}{2}]} = e^{i\alpha} V A^{[S_z=-\frac{1}{2}]} V^\dagger , \tag{21}$$

Now we use the fact that any $2 \times 2$ matrix has a unique expansion in terms of the identity matrix and the Pauli matrices, $\sigma_{x,y,z}$. In our case, we can rewrite $A^{[S_z = \frac{1}{2}]} = 1 + a\sigma_z$ and $A^{[S_z = -\frac{1}{2}]} = \frac{b+c}{2}\sigma_x + \frac{b-c}{2}i\sigma_y$, so the above equation becomes

$$e^{i\frac{\theta}{2}}(1 + a\sigma_z) = e^{i\alpha}V(1 + a\sigma_z)V^\dagger, \tag{22}$$

$$e^{-i\frac{\theta}{2}}\left(\frac{b+c}{2}\sigma_x + \frac{b-c}{2}i\sigma_y\right) = e^{i\alpha}V\left(\frac{b+c}{2}\sigma_x + \frac{b-c}{2}i\sigma_y\right)V^\dagger. \tag{23}$$

Under the conjugation by $V \in SU(2)$, the identity matrix is invariant while the Pauli matrices transform as a vector under $SO(3)$. So, when $a \neq 0$, for the first equation to hold, we must have $e^{i\alpha} = e^{i\frac{\theta}{2}}$ and $V = \exp(i\frac{\sigma_z}{2}\phi)$ for some $\phi \in \mathbb{R}$. Then the second equation becomes

$$e^{-i\theta}\left(\frac{b+c}{2}\sigma_x + \frac{b-c}{2}i\sigma_y\right) = \frac{b+c}{2}(\cos\phi\,\sigma_x + \sin\phi\,\sigma_y) + i\frac{b-c}{2}(-\sin\phi\,\sigma_x + \cos\phi\,\sigma_y)$$

$$= \frac{be^{-i\phi} + ce^{i\phi}}{2}\sigma_x + \frac{be^{-i\phi} - ce^{i\phi}}{2}i\sigma_y,$$

which implies that

$$e^{-i\theta}(b+c) = be^{-i\phi} + ce^{i\phi}, \qquad e^{-i\theta}(b-c) = be^{-i\phi} - ce^{i\phi}. \tag{24}$$

If $bc \neq 0$, then $e^{i\theta} = e^{i\phi} = e^{-i\phi}$, giving us $e^{i\theta} = \pm 1$, i.e., $\theta = 0$ or $\theta = \pi$. Therefore, within the group of $U(1)$ spin rotations with respect to $S_z$, the $\mathbb{Z}_2$ generated by $\pi$ rotation is the only symmetry of the MPS in Eq. (16). Note that if any of $b$ and $c$ vanishes, any $\theta = \phi$ solves these equations and the MPS has a $U(1)$ symmetry corresponding to the $S_z$ rotation.

The above $\mathbb{Z}_2$ symmetry implies that $\langle S_x \rangle = \langle S_y \rangle = 0$ for the MPS in Eq. (16). As long as we can show that $\langle S_z \rangle \neq 0$, we know that the only subgroup of $SO(3)$ spin rotation which can possibly be a symmetry of Eq. (16) must be a subgroup of $U(1)$ spin rotations with respect to $S_z$. Then combined with the previous result, we conclude that the only subgroup of $SO(3)$ spin rotation that is still a symmetry of Eq. (16) is the $\mathbb{Z}_2$ generated by $\pi$ rotation with respect to $S_z$.

In the following, we use the standard transfer matrix method to calculate $\langle S_z \rangle$ for a spin-$1/2$ described by the MPS in Eq. (16), which contains $L$ sites and has periodic boundary condition. To this end, we first consider the tensor

$$M_{i,j,k,l} = \sum_{m,m' = \pm\frac{1}{2}} A_{ij}^{[m]}\hat{S}_{z,mm'}\left(A_{kl}^{[m']}\right)^*, \tag{25}$$

and convert it into a matrix in a way similar to Eq. (18):

$$\tilde{M}_{(ik),(jl)} = M_{i,j,k,l}, \tag{26}$$

where $\hat{S}_z = \text{diag}(1/2, -1/2)$. Explicitly, we have

$$\tilde{M} = \begin{pmatrix} (a+1)^2 & 0 & 0 & -b^2 \\ 0 & 1-a^2 & -bc & 0 \\ 0 & -bc & 1-a^2 & 0 \\ -c^2 & 0 & 0 & (a-1)^2 \end{pmatrix}. \tag{27}$$

Then

$$\langle S_z \rangle = \frac{\text{Tr}(\tilde{M}\tilde{T}^{L-1})}{\text{Tr}(\tilde{T}^L)}. \tag{28}$$

To evaluate the above expression, we perform Jordan decomposition for $\tilde{T}$ and $\tilde{M}$ to obtain $\tilde{T} = S_T J_T S_T^{-1}$ and $\tilde{M} = S_M J_M S_M^{-1}$, where

$$
\begin{aligned}
S_T &= \begin{pmatrix} 0 & 0 & \frac{2a - \sqrt{4a^2 + b^2 c^2}}{c^2} & \frac{\sqrt{4a^2 + b^2 c^2} + 2a}{c^2} \\ -1 & 1 & 0 & 0 \\ 1 & 1 & 0 & 0 \\ 0 & 0 & 1 & 1 \end{pmatrix}, \\
S_M &= \begin{pmatrix} 0 & 0 & \frac{\sqrt{4a^2 + b^2 c^2} - 2a}{c^2} & \frac{-\sqrt{4a^2 + b^2 c^2} - 2a}{c^2} \\ 1 & -1 & 0 & 0 \\ 1 & 1 & 0 & 0 \\ 0 & 0 & 1 & 1 \end{pmatrix},
\end{aligned} \tag{29}
$$

and $J_T = J_M = \mathrm{diag}(1 - a^2 - bc, 1 - a^2 + bc, 1 + a^2 - \sqrt{4a^2 + b^2 c^2}, 1 + a^2 + \sqrt{4a^2 + b^2 c^2})$. Plugging these into Eq. (28) and taking the limit $L \to \infty$ yield

$$
\langle S_z \rangle = \frac{1 + a^2 + \frac{4a^2 - b^2 c^2}{\sqrt{4a^2 + b^2 c^2}}}{1 + a^2 + \sqrt{4a^2 + b^2 c^2}} . \tag{30}
$$

Clearly $\langle S_z \rangle \neq 0$ for generic nonzero $a, b, c \in \mathbb{R}$.

Therefore, we have completed the proof that, for generic nonzero $a, b, c \in \mathbb{R}$, the only subgroup of $SO(3)$ that is still a symmetry of the MPS in Eq. (16) is the $\mathbb{Z}_2$ generated by the $\pi$ rotation with respect to $S_z$, which further implies that the remaining symmetry of the stack of chains described in the main text is $G = \mathbb{Z}_2 \times \mathbb{Z}^d$.

## C.2 $G$-symmetric short-range entangled state beyond the conventional classification

In the previous section, we have already shown that the stack of spin-1/2 chains described in the main text is a $G$-symmetric short-range entangled state. It is then natural to ask how this state fits into the classification of $d + 1$ dimensional $G$ symmetry-protected topological states ($G$-SPTs). In this subsection, we show that this stack of spin-1/2 chains is beyond the classification of $d + 1$ dimensional $G$-SPTs develeped in Refs. [22, 23], and in some sense even beyond the more refined classification proposed in Refs. [18, 24].

According to Refs. [22, 23], the classification of $d + 1$ dimensional $G$-SPTs is $H^{d+1}(G, U(1))$, with $G = \mathbb{Z}_2 \times \mathbb{Z}^d$. This group cohomology has been calculated in Ref. [28], and it is found that

$$
H^{d+1}(G, U(1)) = \prod_{k=1}^{d+1} H^k(\mathbb{Z}_2, U(1))^{\binom{d}{k-1}} . \tag{31}
$$

This classification has a simple interpretation in terms of dimensional reduction: to construct a $G$-SPT in $d$ spatial dimensions, we can first pick up $k - 1$ spatial dimensions, build up $(k - 1) + 1$ spacetime dimensional $\mathbb{Z}_2$-SPTs along these dimensions, and stack these lower dimensional $\mathbb{Z}_2$-SPTs in a way that preserves the $\mathbb{Z}^d$ translation symmetry. Using the fact that $H^k(\mathbb{Z}_2, U(1)) = \mathbb{Z}_2$ for odd $k$ and $H^k(\mathbb{Z}_2, U(1)) = \mathbb{Z}_1$ for even $k > 0$, we can get the classification of $G$-SPTs in various dimensions. When $d = 1, 2$ and 3, the classification is $\mathbb{Z}_2$, $\mathbb{Z}_2^2$ and $\mathbb{Z}_2^4$, respectively. The analysis can be extended to higher $d$, in which case it is still true that our stack of spin-1/2 chains is beyond this group-cohomology-based classification, but below we will focus on the case with $d \leq 3$.

Let us start with the case where $d = 1$ and the classification is $\mathbb{Z}_2$. The dimensional reduction picture implies that the nontrivial $1+1$ dimensional $G$-SPT, roughly speaking, has each of its translation unit cells hosting a $\mathbb{Z}_2$ odd state. More precisely, it means that when the length of the chain increases by 1, the $\mathbb{Z}_2$ eigenvalue of the ground state changes. In other words, the two different $1+1$ dimensional $G$-SPTs are expected to be characterized by $\lambda \equiv \lim_{L \to \infty} \frac{\lambda_{L+1}}{\lambda_L}$, where $\lambda_L$ is the eigenvalue of the ground state with length $L$, $|\psi\rangle_L$, under the $\mathbb{Z}_2$ symmetry action $X$, i.e., $X|\psi\rangle_L = \lambda_L |\psi\rangle_L$ [21]. It is natural to identify the $G$-SPT with $\lambda = 1$ ($\lambda = -1$) as the trivial (nontrivial) SPT. Then a simple example of a trivial (nontrivial) $1+1$ dimensional $G$-SPT is a many-qubit state $\prod_j |0\rangle_j$ ($\prod_j |1\rangle_j$), where the on-site $\mathbb{Z}_2$ symmetry action $X_j$ satisfies $X_j|0\rangle_j = |0\rangle_j$ and $X_j|1\rangle_j = -|1\rangle_j$. Note that both of these SPTs are ultralocal product states, although they have symmetry-protected distinction. This highlights the fact that in the presence of lattice symmetries the notions of trivial and nontrivial SPTs are in some sense semantic.

For our MPS state described by Eq. (16), the results from the previous subsection imply that $\frac{\lambda_{L+1}}{\lambda_L} = e^{i\alpha} = i$, which is neither 1 nor $-1$. This means that this simple MPS is a $G$-symmetric short-range entangled state beyond the conventional group-cohomology-based classification [21], which is possible because in our case the $\mathbb{Z}_2$ on-site symmetry acts on states as a $\mathbb{Z}_4$ group. Notice that this is different from having projective representations under the on-site symmetry ($\mathbb{Z}_2$ actually has no projective representation), so in some sense such a state is not even captured by the more refined classification proposed in Refs. [18, 24]. We also remark that, if only the remaining symmetry $G$ is preserved but $G_0$ is explicitly broken, it should be possible to deform our state such that $b = c = 0$ without closing the gap [21], which smoothly connects our state to the all-spin-up ultralocal product state. During the course of this deformation, $\frac{\lambda_{L+1}}{\lambda_L} = i$ is invariant. Moreover, we can consider another MPS state obtained by switching $A^{[S_z = \frac{1}{2}]} \leftrightarrow A^{[S_z = -\frac{1}{2}]}$, which has $\frac{\lambda_{L+1}}{\lambda_L} = -i$ and can be smoothly connected to the all-spin-down ultralocal product state while preserving the symmetry $G$. This again reflects the fact that the notions of trivial and nontrivial SPTs are somewhat semantic in the presence of lattice symmetries. More generally, it is more appropriate to view these states as being described by a torsor over $H^2(\mathbb{Z}_2 \times \mathbb{Z}, U(1))$, rather than elements of $H^2(\mathbb{Z}_2 \times \mathbb{Z}, U(1))$.

Next, we move to the case with $d = 2$, where the classification is $\mathbb{Z}_2^2$. It is easy to see that one of the two root SPTs is simply the nontrivial SPT protected only by $\mathbb{Z}_2$, where the translation symmetry is unimportant, and the other root SPT is simply the higher dimensional generalization of the $1+1$ dimensional $G$-SPT discussed above. Roughly speaking, each translation unit cell in this SPT hosts a $\mathbb{Z}_2$ odd state. Clearly, our stack of spin chains is neither of these two states, which means it is beyond the group-cohomology-based classification.

Finally, we turn to the case with $d = 3$, where the classification is $\mathbb{Z}_2^4$. One of the four root states is simply again the generalization of the $1+1$ dimensional state, where each translation unit cell hosts a $\mathbb{Z}_2$ odd state. The other three root states can be viewed as stacks of $2+1$ dimensional $\mathbb{Z}_2$-SPTs that have $\mathbb{Z}^3$ translation symmetry. Again, it is clear that our stack of spin chains is none of these states, and it is therefore beyond the group-cohomology-based classification. Instead, such a state should be thought of as a torsor over $H^2(\mathbb{Z}_2 \times \mathbb{Z}, U(1))$, rather than an element in $H^2(\mathbb{Z}_2 \times \mathbb{Z}, U(1))$.

## C.3 Properties of the Goldstone modes

In this subsection, we discuss the properties of the Goldstone modes associated with the $G_0 \to G$ symmetry-breaking orders. This discussion applies to all $S \geq 1/2$.

We will use the results in Ref. [49], which connect the number of broken generators of a continuous symmetry and the expectation values of the charge density in the ground states to the number and dispersion of the Goldstone modes. The expectation values of the charge

density enter through the anti-symmetric matrix $\rho$ defined as

$$\rho_{ab} \equiv -i\langle[\hat{S}_a, \hat{S}_b]\rangle/V \,, \tag{32}$$

where $\hat{S}_{a,b}$ are the broken generators, and $V$ is the volume of the system. Denote the number of the broken generators by $n_{BG}$, Ref. [49] shows that the total number of Goldstond modes is $n_{BG} - \frac{1}{2}\mathrm{rank}(\rho)$, where $n_{BG} - \mathrm{rank}(\rho)$ of them generically have linear dispersion, and the other $\frac{1}{2}\mathrm{rank}(\rho)$ of them generically have quadratic dispersion.

In our case, the number of broken generators is $n_{BG} = 3$. Because the remaining symmetry of $G$ does not require $\langle S \rangle = 0$, the matrix $\rho$ generically has rank 2. So there will be $n_{BG} - \frac{1}{2}\mathrm{rank}(\rho) = 3 - 2/2 = 2$ gapless Goldstone modes, where one of them generically has linear dispersion and the other has quadratic dispersion.

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
