# Peer review of "Entanglement-enabled symmetry-breaking orders"

_SciPost Physics Core, doi:SciPost Phys. Core 7, 010 (2024)_

## Round 1 · Referee Report · Anonymous (Referee 1) · 2023-6-8

Strengths

Interesting idea about classifying symmetry breaking.

Weaknesses

Not written very clearly.

Report

The manuscript "Entanglement-Enabled Symmetry-Breaking Order" seems
interesting to me. As I understand it, the strategy is to first
specify a symmetry of a full hilbert space (or define a symmetry),
then consider how this symmetry might break. If the broken symmetry
can never be a local product state, then it is declared to be a EESBO.

What is not clear to me is whether this manner of classifying broken
symmetries is going to turn out to be useful or not.

There appears to be no requirement that the state in question is
gapped, or even that it is the ground state of any Hamiltonian. This
work is simply making statements about wavefunctions with some
symmetry that live within a Hilbert space with some symmetry. If this
is so, is this a feature or a bug? Does it suggest that the
classification may end up classifying many things that are not phases
of matter in any sense?

Overall I nonetheless think the paper is interesting and should be
published. I'm not sure there will be overwhelming interest, so I
suggest SciPost core rather than the flagship SciPost.

To give some suggestions, I do have to say that the paper was
extremely hard for me to read. While there is a small community who
is very familiar with these types of arguments, I fear that most even
well-educated condensed matter theorists will find much of the
arguments to be very obtuse.

The examples seem needlessly difficult to think about. For example,
section 5.2 seems the simplest example (at least to me) because one
can take a limit fo a=b=c=1 and then the wavefunction is super-easy to
describe and you can just look at it and see what is going on. Trying
to do it in generality makes it completely impossible to understand.
Similarly, the entire example in section 4 is insanely arcane.

Even in section 5, the authors insist on doing a 3d example,
presumably to evade Mermin-Wagner. But there is no point in this.
Since we didn't ever specify a Hamiltonian, why not just assume it is
a long-ranged Hamiltonian, so that Mermin-Wagner does not apply.
(Indeed, the Hamiltonian doesn't matter anyway!) Then you can just
talk about spin chains (Am I wrong about this?).

I would recommend that before publication, the authors try very hard
to simplify much the discussion, clarify the writing, and put all the
simple examples up front. Yes, I know this is hard to do, but it
really will pay off in the end. If you write a paper that only a tiny
fraction of the community can bother to understand, then it won't have
much impact.

Also note, the paragraph where the EESBO is defined in section 2 gives the
definition almost as a side comment. One can read the paragraph and
not even realize that there has just been a definition made. Please
state it clearly and precisely, not so casually, so people know what
you are talking about! And flag it clearly

"DEFINITION: GIven X,Y, Z, we say that a wavefunction is EESBO if P,
Q, R"

Requested changes

See report. Please simplify examples and write more clearly for a bigger audience.

  • validity: high
  • significance: good
  • originality: good
  • clarity: ok
  • formatting: perfect
  • grammar: excellent

Author:  Cheng-Ju Lin  on 2023-10-30  [id 4079]

(in reply to Report 1 on 2023-06-08)

We thank Referee for the valuable report. We have prepared the response in the following PDF file.

Sincerely,
Cheng-Ju Lin and Liujun Zou

Attachment:

Response_GCU1AoI.pdf

---

## Round 1 · Referee Report · Anonymous (Referee 2) · 2023-7-9

Strengths

- Clear and simple
- Interesting idea

Weaknesses

- Not application found
- Missing a classification

Report

The paper is interesting but I think SciPost core fits more for the publication.

The definition of EESBO should be highlight. Maybe talk about the lack of a basis that enable to write the ground states in a product form. Maybe also put an example of not EESBO and long range like the GHZ state from the 2-fold degenerate ground state of the Ising model.

I have some questions about the mechanism of EESBO. The paper considers SPT phases that are not conventional but I think this is not required. Let us consider an ordinary G0= Z2x(ZnxZn) and go to G=ZnxZn; breaking the Z2 symmetry that could be internal and not a lattice symmetry. With the regular SPT classification the 2 ground states could be in n different SPT phases, n-1 of them non-trivial and therefore not able to be written as a product state because of the entanglement given by the projective representation. Why are not those EESBO? If they are, why are they not considered?

I guess that the family of states can be generalized to break T to T^m and have m-degenerate ground states.

Most readers won't be familiar with these unconventional SPT phases. I am familiar with the standard understanding of projective representations at the bonds giving rise to standard SPT phases (also with MPSs) and the RGFP cartoon picture of entangled pairs between sites. Is there an easy characterization or cartoon picture of these unconventional SPT phases? I think it would help a lot if the paper describes them.

Requested changes

See report. Clarify definition and SPT cases

  • validity: good
  • significance: ok
  • originality: good
  • clarity: good
  • formatting: excellent
  • grammar: excellent

Author:  Cheng-Ju Lin  on 2023-10-30  [id 4078]

(in reply to Report 2 on 2023-07-09)

We thank Referee for the valuable report. We have prepared the response in the following PDF file.

Sincerely,
Cheng-Ju Lin and Liujun Zou

Attachment:

Response.pdf

---

## Round 2 · Referee Report · Anonymous (Referee 2) · 2023-11-19

Report

The authors have changed the paper according to my comments.

---

## Round 2 · Referee Report · Anonymous (Referee 3) · 2024-2-7

Report

This paper tries to propose the idea of "Entanglement-enabled symmetry-breaking orders" with several examples. The first example is the same as that discussed in SciPostPhys.11.2.024. In that paper, the fact that there is no simple product state satisfying all the symmetries is presented as an example of the "generalized Lieb-Schultz-Mattis theorem", while in this paper, the authors tried to argue that this example is not covered by the LSM theorem. Of course, one may say that this is a matter of definition, but this example has a lot of similarities with the original LSM example. I tend to agree with the author of SciPostPhys.11.2.024 and consider this example as a generalized LSM. The second example is interesting where the authors pointed out that any pure state of a spin 1/2 has a residue U(1) symmetry, therefore no product state of spin 1/2's can break the SO(3) symmetry down to Z2. This is an interesting observation, although this is a fragile phenomenon in the sense that if other half-integer spin representations are allowed, such an obstruction no longer exists. It is therefore important to not only enforce the symmetry but also enforce the symmetry representation on each lattice site. The last example has been discussed in Phys. Rev. B 94, 064432 as a "featureless quantum insulator" without a tensor product wave-function. So this example is again known.

This paper grouped several examples together, some with a higher level of innovation than others. On the other hand, the paper did not present a unified mechanism that explains all these examples and predicts new ones. The examples are kind of ad hoc. Because of this, I agree with the referees that the level of innovation in this paper is low. Some of the examples are interesting and the paper may be worth publishing because of that.
  • validity: high
  • significance: ok
  • originality: low
  • clarity: high
  • formatting: excellent
  • grammar: excellent

Author:  Cheng-Ju Lin  on 2024-02-18  [id 4315]

(in reply to Report 2 on 2024-02-07)
Category:
remark

We thank Referee for the comments in the report. We have added the last paragraph in Sec. 4, clarifying the difference between our construction and the construction in Ref.[18] (Jiang et. al.). In particular, our construction has spontaneous symmetry breaking while Ref. [18] does not. Moreover, our diagnostic of EESBO does not use the property of short-range entangled states having nontrivial edge modes.

We thank Referee's comment again so that we can clarify the difference between our construction and the construction in Ref.[18] in the main text.

---

## Round 2 · Referee Report · Anonymous (Referee 1) · 2024-2-14

Report

I'm a bit disappointed that the authors chose to not make many changes in the paper. Nonetheless, as I said in my first report, I think the work is worth publishing in SciPost core.

---

## Round 2 · List of Changes

Along with various minor changes, here is a summary of the major changes.
1. We have highlighted the defintion of EESBO.
2. We have added a sentence in Sec. 2 to emphasize that we are considering ground states of local Hamiltonians.
3. We have added a sentence at the end of Sec. 2 to emphasize the difference between entanglement-enabled symmetry-breaking orders and the usual phenomenon of coexistence of spontaneous symmetry breaking and nontrivial topological phases.
4. We have added a sentence in Sec. 5 to provide further insights to the example there.

---

## Round 3 · List of Changes

• Added the last paragraph in Sec. 4, clarifying the difference of our construction from the construction considered in Ref.[18] (Jiang et. al.).

---

## Editorial Decision

published